# Use of Remdesivir in Patients with SARS-CoV-2 Pneumonia in a Real-Life Setting during the Second and Third COVID-19 Epidemic Waves

**DOI:** 10.3390/v15040947

**Published:** 2023-04-11

**Authors:** Raffaella Marocco, Cosmo Del Borgo, Eeva Tortellini, Silvia Garattini, Anna Carraro, Daniela Di Trento, Andrea Gasperin, Alessandra Grimaldi, Tiziana Tieghi, Valeria Belvisi, Blerta Kertusha, Mariasilvia Guardiani, Paola Zuccalà, Danilo Alunni Fegatelli, Alessandra Spagnoli, Miriam Lichtner

**Affiliations:** 1Infectious Diseases Unit, Santa Maria (SM) Goretti Hospital, Sapienza University of Rome, 04100 Latina, Italy; 2Department of Public Health and Infectious Diseases, Sapienza University of Rome, 00185 Rome, Italy; 3Department of Neurosciences, Mental Health, and Sense Organs, NESMOS, Sapienza University of Rome, 00189 Rome, Italy

**Keywords:** remdesivir, SARS-CoV-2, antiviral therapy, COVID-19, real-life study

## Abstract

In this retrospective comparative study, we evaluated the effectiveness of remdesivir (RDSV) in patients with SARS-CoV-2 pneumonia. Individuals hospitalized between March 2020 and August 2022 at S.M. Goretti Hospital, Latina, with a positive test for SARS-CoV-2 and, concomitantly, pneumonia, were included. The overall survival was the primary endpoint. The composite secondary endpoint included death or progression in severe ARDS at 40 days. The study population was stratified according to treatment into two groups: the RDSV group (patients treated with RDSV-based regimens) and the no-RDSV group (patients treated with any other, not RDSV-based, regimens). Factors associated with death and progression to severe ARDS or death were assessed by multivariable analysis. A total of 1153 patients (632 belonging to the RDSV group and 521 to the no-RDSV group) were studied. The groups were comparable in terms of sex, PaO2/FiO2 at admission, and duration of symptoms before hospitalization. Further, 54 patients (8.5%) in the RDSV group and 113 (21.7%) in the no-RDSV group (*p* < 0.001) died. RDSV was associated with a significantly reduced hazard ratio (HR) of death (HR, 0.69 [95% CI, 0.49–0.97]; *p* = 0.03), compared to the no-RDSV group, as well as a significantly reduced OR of progression in severe ARDS or death (OR, 0.70 [95% CI 0.49–0.98]; *p* = 0.04). An overall significantly higher survival rate was observed in the RDSV group (*p* < 0.001, by log-rank test). These findings reinforce the survival benefit of RDSV and support its routine clinical use for the treatment of COVID-19 patients.

## 1. Introduction

The current COVID-19 pandemic, which originated in December 2019, has posed enormous healthcare challenges around the world. It has led to increased hospitalizations for pneumonia with multiorgan disorders and deaths, with new estimates from the World Health Organization (WHO) of 14.9 million excess deaths associated with the pandemic in 2020 and 2021 [1,2,3].

Several therapies are available for hospitalized patients with moderate or severe COVID-19. Some treatments, such as monoclonal antibodies, have proven to reduce the risk of progression of COVID-19, and, in the health emergency, several therapeutic agents have been evaluated for the prevention and treatment of this disease, but the definition of an efficacious drug is still challenging [1,4,5,6].

At the beginning of the pandemic, with no definitive curative treatment insight and a high mortality rate in vulnerable populations, health authorities sought to re-stratify risks and focus on the repurposing of available drugs to develop timely and cost-effective therapeutic strategies, targeting hospitalized and critically ill patients [6,7].

Several antiviral/antimalarial agents such as remdesivir (RDSV), ritonavir/lopinavir combination, hydroxychloroquine, chloroquine, and immuno-modulating therapies such as tocilizumab, sarilumab, convalescent plasma, and interferon were under randomized controlled trials (RCT) in many countries in order to evaluate their efficacy and safety in the treatment of COVID-19 [4,8,9].

RDSV (also known as GS-5734), developed by Gilead Sciences via collaboration with the U.S. Centers for Disease Control and Prevention (CDC) and the U.S. Army Medical Research Institute of Infectious Diseases (USAMRIID) was proposed as a potential candidate drug for the treatment of COVID-19, and recent studies have shown promising results, suggesting that it could represent a ‘molecule of hope’ for the treatment of COVID-19 [4,7,10,11]. 

RDSV is a nucleotide analog prodrug that inhibits viral RNA-dependent RNA polymerase, which was first developed to treat Ebola and further demonstrated an in vitro inhibitory activity against coronaviruses, including SARS-CoV-2 [2,12,13,14,15,16]. Furthermore, early RDSV treatment showed its efficacy in in vivo studies on SARS-CoV-2-infected macaques [17]. 

Several studies indicate how it shortens the recovery time of hospitalized patients [12,13,14,15,16]. In particular, the phase 3 trial of RDSV showed that both a 10-day course and a 5-day course shortened the recovery time in patients hospitalized with COVID-19 [6,8,18]. 

On 10 April 2020, AIFA authorizes the Solidarity study promoted by the WHO, in which different therapeutic strategies were evaluated, including antiviral RDSV and lopinavir/ritonavir alone or in combination with beta interferon, chloroquine, and hydroxychloroquine [9].

Based on these findings, in June 2020, RDSV was proposed by the EMA as the first antiviral drug approved for the treatment of COVID-19 disease [19]. Finally, on 22 October 2020, RDSV became the first United States Food and Drug Administration (FDA)-approved drug for the treatment of hospitalized COVID-19 patients [11,19]. In Italy, RDSV has been available since October 2020.

Here, we report a retrospective study that aimed at evaluating the effectiveness of RDSV-based regimens on hospitalized patients with SARS-CoV-2 pneumonia, using mortality as the primary endpoint and mortality or progression to severe ARDS as the secondary endpoint in univariate and multivariate models in a real-life context.

## 2. Materials and Methods

### 2.1. Ethics Statement

This study was approved by the Ethics Committee Lazio 2 (protocol number 0038491/2022), as established by the Ministry of Health of the Italian Government. Each subject gave written informed consent for data analysis.

### 2.2. Study Design and Patients

A retrospective comparative study was conducted on hospitalized patients between March 2020 and August 2022 at the S.M. Goretti Hospital, Latina.

Inclusion criteria were an age of >18 years, confirmed SARS-CoV-2 infection by polymerase chain reaction assay (PCR), and concurrent pneumonia.

The study population included patients treated with RDSV-based regimens (RDSV group) and patients treated with any other, not RDSV-based, regimens (no-RDSV group).

Specifically, patients belonging to the no-RDSV group met the exclusion criteria for RDSV or were hospitalized before October 2020. Exclusion criteria for RDSV use included alanine aminotransferase (ALT) or aspartate aminotransferase (AST) levels greater than 5 times the upper limit of the normal range, estimated creatinine clearance less than 30 mL per minute (by the Cockcroft–Gault formula), or duration of symptoms more than 10 days.

During the study period, the most common viral variants in Italy were the wild-type Wuhan variant, the Delta variant from 2 June 2021 to 15 December 2021, and Omicron from 16 December 2021 [20].

RDSV was administered intravenously in patients with radiologic evidence of pneumonia and oxygen support, following the national guidelines available at the time of hospitalization. According to local protocol, all the case patients received 200 mg of RDSV on day 1 as a loading dose, followed by 100 mg once daily for the subsequent 4 days as a maintenance dose, for a total of 5 days of treatment. 

RDSV-based regimens consisted of RDSV plus standard of care (SOT), including steroids, enoxaparin, and tocilizumab, whereas controls included SOT ± lopinavir/ritonavir/hydroxychloroquine. The use of steroids or tocilizumab was based on clinical judgment and on the national and local guidelines available at that time.

A description of the general population enrolled was provided, considering variables such as sex distribution, age, symptoms presented at admission, PaO2/FiO2 ratio at admission and the nadir, blood test values (white blood cell count, percentage of lymphocytes and neutrophils, creatinine, C-reactive protein (CRP), vaccination status, comorbidities, acute respiratory distress syndrome (ARDS) and its grade (low, moderate, or severe), and death during hospitalization.

The study population was stratified according to treatment into two groups: the RDSV group (patients treated with RDSV-based regimens) and the no-RDSV group (patients treated with any other, not RDSV-based, regimens), and the differences between them were evaluated.

The effects of RDSV-based regimens were investigated using mortality as the primary endpoint and mortality or progression in severe ARDS as a composite endpoint.

In order to better understand the role of antivirals, an ulterior stratification was performed in the no-RDSV group, dividing it into Lop/rit/chloro (patients receiving either lopinavir/ritonavir/chloroquine) and Other (any other, not RDSV-based, or lopinavir/ritonavir/chloroquine regimens).

### 2.3. Statistical Analysis

The data are reported as medians with interquartile ranges (IQR) (25th–75th percentile) for continuous variables and as frequencies and percentages for categorical variables. Baseline characteristics were compared by the Kruskal–Wallis test or Chi-Square, as appropriate. 

Overall survival (OS) was defined as the time from the date of hospitalization to death, from any cause. Patients who do not have OS events were censored at the date they were last known to be alive.

The primary endpoint is defined as the time from hospitalization to the first documentation of death. Patients who do not have death events will be censored at their last disease assessment date.

The composite endpoint is defined as the time from hospitalization to the first documentation of progression in severe ARDS or death. Patients who do not have progression in severe ARDS or death events will be censored at their last disease assessment date.

The OS probabilities were estimated in each group using the non-parametric Kaplan–Meier method and displayed graphically. The groups’ differences in OS were assessed by log-rank test. To estimate the association between RDSV use and mortality, multivariable Cox proportional hazard regression analyses were applied. The proportional hazards assumption was verified using graphical methods; scaled Schoenfeld residuals and graphical checks proposed by Klein and Moeschberger were performed. 

A multivariable logistic regression model was used to assess the influences of covariates on the composite endpoint. 

The *p*-values < 0.05 were considered significant. The confidence intervals were at a 95% level. All analyses were performed using the software R (version 4.2.2 R Foundation for Statistical Computing, Vienna, Austria).

## 3. Results

### 3.1. Demographic and Clinical Characteristics of the Study Population

A total of 1153 patients with confirmed SARS-CoV-2 infection were included in the study and systematically followed up during hospital stay.

Specifically, 632 patients were treated with RDSV based-regimens (RDSV group) and 521 were treated with any other, not RDSV-based, regimens (no-RDSV group); among them, 160 received lopinavir/ritonavir/ chloroquine based-regimens (Lop/rit/chloro subgroup) and 361 did not receive any antiviral at all (Other subgroup) (Figure 1). 

Demographic and clinical characteristics of the study population at hospital admission are shown in Table 1. In detail, 458 were females and 695 males with a median age (IQR) of 65 years [53–76] (Table 1). Overall, the median (IQR) PaO2/FiO2 at admission was 309 [243–366] and the median (IQR) duration from symptoms onset to hospitalization was 6 days [3–10] (Table 1). The two groups were comparable in terms of sex, PaO2/FiO2 at admission, and duration of symptoms before hospitalization (Table 1).

Conversely, as expected, in the RDSV group, age (*p* < 0.001) and creatinine (*p* < 0.001) were lower compared to the no-RDSV group (Table 1).

Furthermore, a lower proportion of patients with comorbidities was found in the RDSV group compared to the counterpart (*p* = 0.017). Types of comorbidities were similar, with the exception of CRF, cardiovascular diseases, and dementia (*p* < 0.001, *p* < 0.001, and *p* < 0.001, respectively) (Table 1).

Among all patients, 981 contracted the infection during the dominance of the Wuhan original variant, 104 during the dominance of the delta variant period, and 68 during the omicron era and no differences were found in terms of vaccination status (Table 1).

### 3.2. Clinical Progression of the Study Population

As reported in Table 2, during hospitalization, 167 patients died. Among them, 113 (21.7%) belonged to the no-RDSV group and 54 (8.5%) to the RDSV group, revealing significantly lower mortality in the latter (*p* < 0.001) (Table 2). 

Concerning progression in ARDS, no differences were found between the two groups (Table 2).

A lower hospital stay was observed in the RDSV group with respect to the no-RDSV group (*p* = 0.017) (Table 2).

Regarding the analysis of adverse effects, although no serious adverse events were reported, four patients voluntarily interrupted RDSV-based regimens complaining of manifestations of allergy and tachycardia.

### 3.3. Survival Analysis

Overall, at 40 days, a survival probability (IQR) of 0.653 [0.584–0.712] was observed for all patients.

Stratifying the study population into the RDSV and no-RDSV groups, survival probabilities of 0.681 [0.556–0.778] and 0.616 [0.537–0.686] were observed in the RDSV group and no-RDSV group, respectively, revealing a significantly higher survival rate for the RDSV group (*p* < 0.001, by the log-rank test) (Figure 2A). 

By Cox proportional hazard regression analysis, the RDSV regimen was associated with a reduced hazard ratio (HR) of death, (HR, 0.69 [95% CI, 0.49–0.97]; *p* = 0.03), together with PaO2/FiO2 at admission (HR, 0.60 [95% CI 0.51–0.71]; *p* < 0.001), whereas classical factors such as age (HR, 2.11 [95% CI 1.82–2.45]; *p* < 0.001), and comorbidities (HR, 1.50 [95% CI 0.86–2.63]) were associated with an increased HR of death (although the latter was not significant) (Figure 2B).

These results were confirmed in the second stratification. On stratifying the no-RDSV group into Lop/rit/chloro and Other subgroups, survival probabilities of 0.735 [0.624–0.864] and 0.555 [0.470–0.655] were observed, respectively, with a statistically significant difference in the OS between the three groups (*p* < 0.001, by the log-rank test) (Figure 2C).

By Cox proportional hazard regression analysis, RDSV (HR, 0.63 [95% CI, 0.44–0.90]; *p* = 0.01) was confirmed to be a protective factor, as was PaO2/FiO2 at admission (HR, 0.61 [95% CI 0.51–0.72]; *p* < 0.001) (Figure 2D). Furthermore, the Lop/rit/chloro (HR, 0.65 [95% CI, 0.39–1.08]; *p* = 0.10) regimen was found to be associated with a reduced HR of death, although not significant, whereas age (HR, 2.08 [95% CI 1.79–2.41]; *p* < 0.001) maintained its role in predicting the death endpoint (Figure 2D).

### 3.4. Analysis of Composite Endpoint

In the multivariable logistic analysis, factors such as age (odds ratio (OR), 1.66 [95% CI 1.45–1.92]; *p* < 0.001) and comorbidities (OR, 1.60 [95% CI 1.00–2.61]; *p* = 0.05) were associated with progression to severe ARDS or death, whereas the RDSV regimen was associated with a reduced odds to develop the composite endpoint (OR, 0.70 [95% CI 0.49–0.98]; *p* = 0.04) together with lymphocytes (OR, 0.96 [95% CI 0.94–0.98]; *p* < 0.001) (Figure 3A).

Performing the same analysis stratifying no-RDSV into the Lop/rit/chloro and Others subgroup, age (odds ratio (OR), 1.66 [95% CI 1.45–1.92]; *p* < 0.001) and comorbidities (OR, 1.60 [95% CI 1.00–2.61]; *p* = 0.05) maintained their role in predicting the composite endpoint, while the RDSV regimen confirmed in reduced OR of death or progression to severe ARDS (OR, 0.69 [95% CI, 0.48–1.00]; *p* = 0.05) together with lymphocytes (OR, 0.96 [95% CI 0.94–0.98]; *p* < 0.001) (Figure 3B). Lop/rit/chloro regimen resulted to be associated with a reduced OR, although not significant (OR, 0.96 [95% CI, 0.54–1.70]; *p* = 0.90) (Figure 3B).

## 4. Discussion

Here, we report the results of a retrospective comparative study aimed at evaluating the effectiveness of RDSV on hospitalized patients with SARS-CoV-2 pneumonia, using overall survival as the primary endpoint and death or progression in severe ARDS as the composite secondary endpoint. Overall, RDSV use within 10 days represented a protective factor associated with a statistically significant decrease in mortality or risk to develop a severe ARDS. Furthermore, a higher survival rate was observed in the RDSV-treated group compared to those who did not receive RDSV, reinforcing the survival benefit of this antiviral and supporting its clinical routine use for the treatment of COVID-19 patients.

Starting from December 2021, RDSV was the only COVID-19 treatment that had received full FDA approval. However, questions remain about its real-world effectiveness. 

Currently, RDSV’s effectiveness in preventing deaths is still debated. Results of clinical trials and observational studies are not always in accordance, ranging from no survival benefit to significant mortality reduction.

The WHO international COVID-19 guidelines recommended against its use, and after that the WHO Solidarity open-label trial showed no effects on the mortality rate among patients [21]. Conversely, other studies reported survival benefits in a subset of patients, including the pivotal adaptive COVID-19 Treatment Trial 1 (ACTT-1) and propensity score matching (PSM) study conducted in the United States, which demonstrated improved clinical recovery in a certain subset of patients receiving RDSV and numerically lower inpatient mortality [21,22,23,24]. Several other real-world studies showed a significant survival benefit of this antiviral for all patients [23,24,25,26,27,28].

In order to evaluate the effectiveness of RDSV-based regimens, our study population was stratified according to treatment into the RDSV group and the no-RDSV group. The two groups were comparable in terms of sex, PaO2/FiO2 at admission, and duration of symptoms before hospitalization, and the statistically significant higher age and creatinine levels found in the no-RDSV group, as well as the higher proportion of patients with comorbidities, is consistent with the fact that patients belonging to the no-RDSV group were often older people with impaired kidney function who did not meet the inclusion criteria for RDSV.

Our study, conducted on 1153 hospitalized patients, reinforces the survival benefit of RDSV. In fact, in multivariate analysis, RDSV-based regimens represented a protective factor associated with a statistically significant decrease in mortality or risk to develop a severe ARDS. It should be noted that very few patients had been vaccinated at the time of hospital admission, suggesting that the benefits observed were driven at least in part by RDSV. Furthermore, most of the enrolled patients were admitted to the hospital during the predominance of wild-type Wuhan and pre-Delta strains, which were known to cause severe disease, especially among unvaccinated persons, supporting its potential beneficial role for all variants of the virus that are less pathogenic. 

Exploring the risk factors related to the endpoints of interest in our population, we found that classical factors such as higher age and the presence of comorbidities were associated with mortality or the risk to develop severe ARDS, while factors such as female sex and higher PaO2/FiO2 at admission resulted to be protective, in accordance with other studies [29,30,31]. These data confirm that the elderly and patients with comorbidities represent a vulnerable population to the worse outcome of COVID-19 and that the mortality of RDSV-treated patients is significantly related to the need for higher levels of O_2_ support. Furthermore, a lymphocyte count in a normal range was found to be associated with reduced mortality or the risk to develop severe ARDS, in accordance with the fact that lymphopenia represents a predictor of poor prognosis in COVID-19 patients, together with older age and comorbidities [32,33,34].

These findings are also consistent with studies that demonstrated the efficacy of RDSV use within nine days from symptom onset in reducing in-hospital mortality, probably due to the kinetics of the viral load of SARS-CoV-2 in the respiratory tract [30].

An ulterior analysis was performed stratifying the no-RDSV group in Lop/rit/chloro and Other subgroups in order to better understand the role of antivirals. This analysis confirmed the beneficial role of RDSV-based regimens and showed that Lop/rit/chloro regimen also was associated with a decrease in mortality compared to therapeutic regimens that did not include any antivirals at all.

In the Omicron era, in which most of the population has developed a natural, and vaccine or booster-delivered immunity, the effectiveness of many antivirals, such as RDSV, remains unclear. Hence, we think it would be useful to conduct and improve randomized trials involving vaccinated patients infected by Omicron variant [35]. 

This is of particular relevance since a newly published multicenter randomized controlled study that aimed at evaluating the efficacy and safety of Paxlovid in hospitalized adult patients with Omicron variant infection showed no significant reduction in the risk of all-cause mortality on day 28 and the duration of SARS-CoV-2 RNA clearance in hospitalized adult COVID-19 patients with severe comorbidities, in discordance with the results obtained previously [36]. Paxlovid consists of two active principles: nirmatrelvir, which is an inhibitor of the SARS-CoV-2 3-chymotrypsin-like cysteine protease enzyme, and ritonavir, a protease inhibitor. It can be dispensed at community pharmacies, and, with respect to remdesivir, which needs to be administered intravenously, Paxlovid has the advantage to be administered orally. Currently, it is recommended by the WHO guidelines only for the treatment of mild-to-moderate COVID-19 and not for COVID-19 pneumonia, underlining the need for a safe and effective drug against severe COVID-19 [37,38].

To our knowledge, there are no comparative trials between RDSV and Paxlovid. Only a study conducted on symptomatic adults hospitalized with mild-to-moderate COVID-19 that compares a five-day course with an oral derivate of RDSV to Paxlovid reports the same effectiveness between the two antivirals with respect to the time to sustained clinical recovery, with fewer safety concerns for the oral derivate of RDSV [38].

However, a recent phase 3 trial conducted in a predominantly vaccinated population infected with various SARS-CoV-2 variants of concern showed the efficacy of a single subcutaneous dose of pegylated interferon lambda administered within seven days after the onset of symptoms in reducing the risk of hospitalization, giving hope for the identification of convenient, widely available, and effective antiviral therapies against COVID-19 [39].

In our study population, RDSV was well tolerated, since no severe adverse events have been reported. This is in line with recent studies that provided insight into the safety of RDSV in the context of the COVID-19 pandemic and demonstrated no significant harm with its use. In fact, the adverse events described are similar across all studies and may suggest that these could be a result of COVID-19 severity rather than the RDSV treatment [4].

This study has several limitations, including the monocentric and retrospective nature, which does not permit generalizations about the results obtained. Furthermore, it includes a low number of vaccinated patients infected during the Omicron era.

Despite these limitations, we believe that the study adds to the body of knowledge on the use of RDSV in real-world settings, considering the large size of the population, which represents a great point of strength. 

In summary, this retrospective study showed that RDSV-based regimens represented a protective factor that did improve mortality overall and was associated with a statistically significant increased likelihood of survival, supporting their use in hospitalized COVID-19 patients and adding another option to the armamentarium for the treatment of patients who are at high risk to develop severe COVID-19.

## Figures and Tables

**Figure 1 viruses-15-00947-f001:**
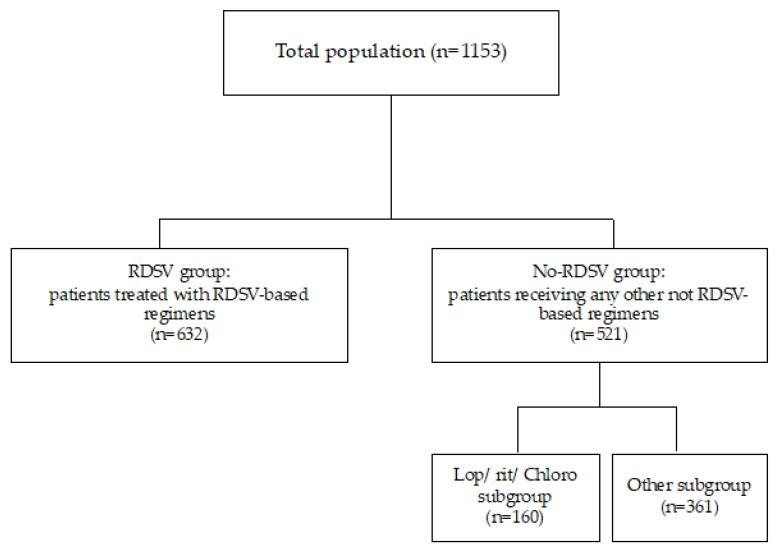
Flow-chart of study population.

**Figure 2 viruses-15-00947-f002:**
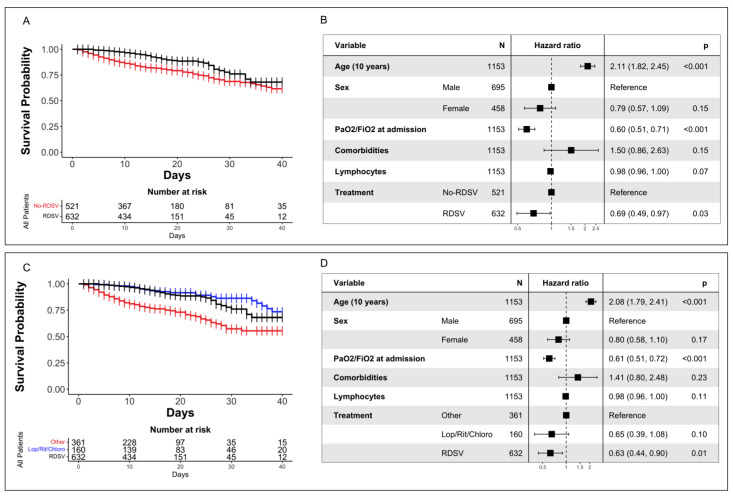
Analysis of primary endpoint. (**A**). Kaplan–Meier survival curve in the RDSV and no-RDSV groups. (**B**) Multivariable Cox analysis of factors associated with the primary endpoint in RDSV and no-RDSV groups. (**C**). Kaplan–Meier survival curve in RDSV, Lop/rit/chloro, and Other. (**D**) Multivariable Cox analysis of factors associated with the primary endpoint in RDSV, Lop/rit/chloro, and Other.

**Figure 3 viruses-15-00947-f003:**
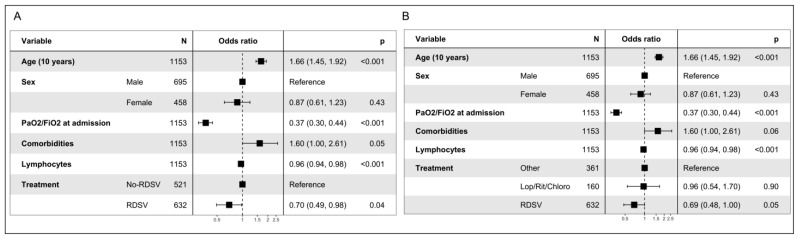
Multivariable analysis of factors associated with the composite endpoint.

**Table 1 viruses-15-00947-t001:** Demographic and clinical characteristics of the study population.

Variable	All	No-RDSV Group	RDSV Group	*p*
n	1153	521	632	
Sex (f/m) (n, %)	458/695	221 (42.4)/300 (57.6)	237 (37.5)/395 (62.5)	0.101
Age	65.00 [53.00–76.00]	68.00 [55.00–81.00]	62.00 [53.00–73.00]	<0.001
PaO2/FiO2 at admission	309.00 [243.00–366.00]	311.00 [233.00–372.00]	308.00 [251.75–361.00]	0.919
PaO2/FiO2 nadir	194.00 [118.00–316.00]	196.00 [116.00–322.00]	191.50 [120.00–312.00]	0.879
Neutrophils	76.50 [66.70–84.50]	76.80 [65.60–85.20]	76.10 [67.18–83.90]	0.454
Lymphocytes	15.40 [9.10–23.00]	14.70 [8.60–23.40]	15.75 [9.70–22.60]	0.568
t_hospitalization symptoms	6.00 [3.00–10.00]	7.00 [3.00–11.00]	6.00 [4.00–9.00]	0.088
Glycemia_mg_dl_.1	112.00 [97.00–141.00]	112.00 [97.00–143.00]	112.00 [98.00–137.00]	0.627
Creatinine_mg_dl_.1	0.88 [0.76–1.09]	0.93 [0.77–1.27]	0.84 [0.75–1.01]	<0.001
CRP_mg_dl_.1	4.39 [1.47–9.60]	4.68 [1.42–10.78]	4.05 [1.48–8.38]	0.176
**Comorbidities** (n, %)	808/1153 (70.1)	384/521 (73.7)	424/632 (67.1)	0.017
Hypertension	538/1153 (47.0)	253/521 (48.6)	285/632 (45.1)	0.265
Diabetes	196/1153 (17.0)	100/521 (19.2)	96/632 (15.2)	0.085
CRF	76/1153 (6.5)	62/521 (11.9)	14/632 (2.2)	<0.001
Obesity	95/1153 (8.2)	35/521 (6.7)	60/632 (9.5)	0.110
Cardiovascular diseases	294/1153 (25.5)	168/521 (32.2)	126/632 (19.9)	<0.001
Neoplasia	81/1153 (7.0)	52/521 (10.0)	29/632 (4.6)	0.001
Neurological diseases	17/1153 (1.5)	7/521 (1.3)	10/632 (1.6)	0.929
Respiratory diseases	156/1153 (13.5)	79/521 (15.2)	77/632 (12.2)	0.166
Dementia	56/1153 (4.9)	42/521 (8.1)	14/632 (2.2)	<0.001
Hypothyroidism	55/1153 (4.8)	25/521 (4.8)	30/632 (4.7)	1.000
**Variant-period** (n, %)				<0.001
Wuhan-period (21 February 2020– 1 June 2021)	981/1153 (85.1)	464/521 (89.1)	517/632 (81.8)	
Delta-period (2 June 2021–15 December 2021)	104/1153 (9.0)	27/521 (5.2)	77/632 (12.2)	
Omicron-period (16 December 2021–present)	68/1153 (5.9)	30/521 (5.8)	38/632 (6.0)	
Vaccination status (n, %)	84/1153 (7.3)	29/521 (5.6)	55/632 (8.7)	0.054

Data are shown as median (IQR) or no. (%) of subjects; CRP: C-reactive protein; CRF: chronic renal failure.

**Table 2 viruses-15-00947-t002:** Clinical progression of the study population.

Variable	All	No-RDSV Group	RDSV Group	*p*
n	1153	521	632	
Mortality (n, %)	167/1153 (14.5)	113/521 (21.7)	54/632 (8.5)	<0.001
**ARDS** (n, %)				
No ARDS	319/1153 (29.9)	147/521 (28.2)	172/632 (27.2)	0.524
Mild	230/1153 (19.9)	107/521 (20.5)	123/632 (19.5)	
Moderate	404/1153 (35.0)	171/521 (32.8)	233/632 (36.9)	
Severe	200/1153 (17.3)	96/521 (18.4)	104/632 (16.5)	
Hospital stay (days)	14 [8–21]	14 [8–24]	13 [8–19]	0.017

Data are shown as median (IQR) or no. (%) of subjects; ARDS: acute respiratory distress syndrome.

## Data Availability

The raw data supporting the conclusions of this article will be made available by the authors without undue reservation.

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
