# Peer review of "Use of Remdesivir in Patients with SARS-CoV-2 Pneumonia in a Real-Life Setting during the Second and Third COVID-19 Epidemic Waves"

_viruses, 2023, doi:10.3390/v15040947_

Round 1
Reviewer 1 Report
The present study is a retrospective comparative study reporting the effectiveness of remdesivir in patients affected by SARS-CoV-2 pneumonia. Data demonstrated survival benefit of remdesivir supporting its clinical routine use for the treatment of COVID-19. The manuscript is well-written and results are clearly exposed. I suggest only to improve the quality of figures (Figure 2 and 3) for the acceptance of the article.
Author Response
The present study is a retrospective comparative study reporting the effectiveness of remdesivir in patients affected by SARS-CoV-2 pneumonia. Data demonstrated survival benefit of remdesivir supporting its clinical routine use for the treatment of COVID-19. The manuscript is well-written, and results are clearly exposed. I suggest only to improve the quality of figures (Figure 2 and 3) for the acceptance of the article.
Response: As kindly suggest by the Referee, we improved the quality of Figure 2 and Figure 3 exporting them in a higher resolution and reinserted them in the main text.

Reviewer 2 Report
Thank you for giving me the opportunity to read and comment a report “Use of remdesivir in patients with SARS-CoV-2 pneumonia in a real-life setting during the second and third COVID-19 epidemic waves”, by Marocco R, et al.
In the reviewed manuscript, the effectiveness of RSDSV-based regimens on hospitalized COVID-19 patients, has been evaluated.
This paper is well written, correctly structured with a suitable research concept, the study limitations are addressed, and it is of relevance to readers of the journal. However, I include a few comments for your consideration.
· In subsection 2.1 of the Material and Methods section, the protocol number, and the date of acceptance of the study by the corresponding ethics committee should be included.
· Will suggest including adjusted analyses and reporting of adjusted Hazard ratios and 95% confidence intervals.
· The Discussion section usually begins with a brief summary of the main findings. Therefore, the first paragraph of the Discussion section should be modified.
· Differences in key outcomes based on age, sex, PAO2/Fio2 at admission, Comorbidities, and Lymphocytes were not discussed, which needs to be discussed accordingly in relation to available lit/review.
· There is now another drug approved for COVID-19, Paxlovid. The possible role of Paxlovid compared to remdesevir should be discussed.
Author Response
Thank you for giving me the opportunity to read and comment a report “Use of remdesivir in patients with SARS-CoV-2 pneumonia in a real-life setting during the second and third COVID-19 epidemic waves”, by Marocco R, et al. In the reviewed manuscript, the effectiveness of RSDSV-based regimens on hospitalized COVID-19 patients, has been evaluated. This paper is well written, correctly structured with a suitable research concept, the study limitations are addressed, and it is of relevance to readers of the journal. However, I include a few comments for your consideration.
- In subsection 2.1 of the Material and Methods section, the protocol number, and the date of acceptance of the study by the corresponding ethics committee should be included.
Response 1: As kindly suggested by the Referee, we included the Ethics Committee, the protocol number and the date of acceptance of the study (lines 84-85) in subsection 2.1. of the Material and Methods section.
- Will suggest including adjusted analyses and reporting of adjusted Hazard ratios and 95% confidence intervals.
Response 2: In this case, the multivariable Cox regression models are not adjusted since we have estimated the coefficient of each covariate inserted in the model holding all other independent variables constant.
- The Discussion section usually begins with a brief summary of the main findings. Therefore, the first paragraph of the Discussion section should be modified.
Response 3: As suggested by the Referee, we modified the first paragraph of the Discussion section with a brief summary of the main findings of the study (lines 237-244).
- Differences in key outcomes based on age, sex, PAO2/Fio2 at admission, Comorbidities, and Lymphocytes were not discussed, which needs to be discussed accordingly in relation to available lit/review.
Response 4: As suggested by the Referee, we implemented the paragraph in which we discuss the differences in key outcomes based on age, sex, PAO2/Fio2 at admission, comorbidities, and lymphocytes (lines 274-284). Specifically, we added three articles to the Bibliography (Ziadi et al., 2021; doi: 10.1111/ijlh.13351; Li et al., 2020; doi: 10.1038/s41375-020-0910-1 and Wu et al., 2020; doi:10.1001/jamainternmed.2020.0994). Furthermore, we added two phrases in subsection 3.4. regarding the association between Lymphocytes and the composite endpoint (line 225 and lines 230-231).
- There is now another drug approved for COVID-19, Paxlovid. The possible role of Paxlovid compared to remdesevir should be discussed.
Response 5: To our knowledge, there are no comparative trials between paxlovid and remdesivir. We found only a study conducted on patients with mild-to-moderate COVID-19 that compares the use of an oral derivate of remdesivir to paxlovid, finding the same effectiveness with respect to the time to sustained clinical recovery, with fewer safety concerns (Z. Cao et al.,2022. DOI: 10.1056/NEJMoa2208822). As suggested by the Referee, we implemented the Discussion section adding information about Paxlovid in relation to remdesivir (lines 297-313).

Reviewer 3 Report
The manuscript concerns an effectiveness of remdesivir (RDSV) in patients with SARS-CoV-2 pneumonia, and it is based on data obtained during SARS-CoV-2 pandemic. The Authors realize the limitations of such studies (e.g., its monocentric and retrospective nature that does not allow wider generalizationof the results), nevertheless, in my opinion this study is valuable and it should be published.
The manuscript is well written and carefully prepared. It required only few editorial corrections (e.g., the dot should be placed after –and not before - the references in the line 45; a space inserted before references in the line 49, unnecessary dot in the subtitle 3.4. should be deleted).
Moreover, since the study is mainly based on statistics, to make this manuscript more suitable for the journal Viruses, my suggestion is to add to the introduction or discussion more information on remdesivir (RDSV). The Authors wrote that “RDSV is a nucleotide analogue prodrug that inhibits viral RNA-dependent RNA polymerase which was firstly developed to treat Ebola and further demonstrated an in-vitro inhibitory activity against coronaviruses, including SARS-CoV-2” but maybe more information about its application, mode of use, or side-effects can be added.
Author Response
The manuscript concerns an effectiveness of remdesivir (RDSV) in patients with SARS-CoV-2 pneumonia, and it is based on data obtained during SARS-CoV-2 pandemic. The Authors realize the limitations of such studies (e.g., its monocentric and retrospective nature that does not allow wider generalizationof the results), nevertheless, in my opinion this study is valuable, and it should be published. The manuscript is well written and carefully prepared.
It required only few editorial corrections (e.g., the dot should be placed after –and not before - the references in the line 45; a space inserted before references in the line 49, unnecessary dot in the subtitle 3.4. should be deleted).
Response 1: As kindly suggested by the Referee, we placed the dot after the references in line 45, inserted a space before references in line 49, removed the dot in subtitle 3.4. and checked again all the possible editorial mistakes in order to correct them.
Moreover, since the study is mainly based on statistics, to make this manuscript more suitable for the journal Viruses, my suggestion is to add to the introduction or discussion more information on remdesivir (RDSV). The Authors wrote that “RDSV is a nucleotide analogue prodrug that inhibits viral RNA-dependent RNA polymerase which was firstly developed to treat Ebola and further demonstrated an in-vitro inhibitory activity against coronaviruses, including SARS-CoV-2” but maybe more information about its application, mode of use, or side-effects can be added.
Response 2: As kindly suggested by the Referee, we added some information about remdesivir. In particular we added an article about its efficacy in vivo in the Introduction section (lines 63-65, Williamson et al., 2020; https://doi.org/10.1038/s41586-020-2423-5) and provided more information about the side-effects of the drug in the Discussion section (lines 319-324, Lin et al., 2021; doi: 10.1007/s15010-020-01557-7). Information about its application and mode of use can be found in section Materials and Methods, paragraph 2.2.
